# Can the Mitochondrial Metabolic Theory Explain Better the Origin and Management of Cancer than Can the Somatic Mutation Theory?

**DOI:** 10.3390/metabo11090572

**Published:** 2021-08-25

**Authors:** Thomas N. Seyfried, Christos Chinopoulos

**Affiliations:** 1Department of Biology, Boston College, Chestnut Hill, MA 02467, USA; 2Department of Medical Biochemistry, Semmelweis University, 1094 Budapest, Hungary; chinopoulos.christos@med.semmelweis-univ.hu

**Keywords:** mutations, IDH1, glycolysis, glutaminolysis, mitochondrial substrate level phosphorylation, ketogenic metabolic therapy, metastasis, oncogenes, chimpanzees, fermentation, respiration, evolution

## Abstract

A theory that can best explain the facts of a phenomenon is more likely to advance knowledge than a theory that is less able to explain the facts. Cancer is generally considered a genetic disease based on the somatic mutation theory (SMT) where mutations in proto-oncogenes and tumor suppressor genes cause dysregulated cell growth. Evidence is reviewed showing that the mitochondrial metabolic theory (MMT) can better account for the hallmarks of cancer than can the SMT. Proliferating cancer cells cannot survive or grow without carbons and nitrogen for the synthesis of metabolites and ATP (Adenosine Triphosphate). Glucose carbons are essential for metabolite synthesis through the glycolysis and pentose phosphate pathways while glutamine nitrogen and carbons are essential for the synthesis of nitrogen-containing metabolites and ATP through the glutaminolysis pathway. Glutamine-dependent mitochondrial substrate level phosphorylation becomes essential for ATP synthesis in cancer cells that over-express the glycolytic pyruvate kinase M2 isoform (PKM2), that have deficient OxPhos, and that can grow in either hypoxia (0.1% oxygen) or in cyanide. The simultaneous targeting of glucose and glutamine, while elevating levels of non-fermentable ketone bodies, offers a simple and parsimonious therapeutic strategy for managing most cancers.

## 1. Introduction

Cancer is a systemic disease involving multiple time- and space-dependent changes in the health status of cells and tissues that ultimately lead to malignant tumors [1,2]. Dysregulated cell growth, i.e., neoplasia, is the biological endpoint of the disease [3]. Tumor cell invasion into surrounding tissues and their spread (metastasis) to distant organs is the primary cause of morbidity and mortality of most cancer patients [4,5,6,7]. Data from the American Cancer Society showed that the number of people dying in the US from cancer in 2013 was 580,350, and in 2020 it was 606,520, an increase of 4.3% [8,9]. The US population increase over this same period was 4.5%, indicating no real progress in cancer management. Cancer is predicted to overtake heart disease as the leading cause of death in Western societies. Is the failure to reduce the cancer death rate due to an incorrect theory on the origin of the disease?

## 2. Scientific Theories

A scientific theory is simply an attempt to explain the facts of nature. Reality is based on replicated facts, whereas interpretation of the facts is based on credible theories. The heliocentric theory of Copernicus, Galileo, and Keppler was able to explain better the movements of celestial bodies than was the geocentric theory of Ptolemy. The germ theory of Louis Pasteur was able to explain better the origin of contagious diseases than was the miasma “bad air” theory of Hippocrates and Galen. The Darwin–Wallace theory of evolution by natural selection was able to explain better the origin of species than was the theory of special creation [10]. In none of these examples could a hybrid theory be envisioned. A theory that can best explain the facts of a phenomenon is more likely to advance knowledge than is a theory less able to explain the facts. The provocative question before us is whether the Mitochondrial Metabolic Theory (MMT) can explain better the origin and management of cancer than can the current Somatic Mutation Theory (SMT).

## 3. The Somatic Mutation Theory of Cancer

According to the SMT, cancer is a complex genetic disease that arises from inherited or random somatic mutations in proto-oncogenes or in tumor suppressor genes [11,12,13]. While many mutations have been found in various tumors, the so-called driver gene mutations are considered most responsible for causing the disease [11,14]. Although the SMT is the dominant scientific explanation for the origin of cancer, numerous inconsistencies have emerged that seriously challenge the credibility of this theory [15,16,17]. The major inconsistencies include:

(1) The absence of gene mutations and chromosomal abnormalities in some cancers [17,18,19,20,21]. For example, Greenman et al. found no mutations following extensive sequencing in 73/210 cancers [13], whereas Parsons et al., found no mutations in the P53, the PI3K, or the RB1 pathways in the Br20P tissue sample of a glioblastoma patient [22]. Cancer cells with no mutations should not exist according to the SMT.

(2) The identification and clonal expansion of numerous driver gene mutations in a broad range of normal human tissues [23,24,25,26,27]. If driver genes cause cancer according to the SMT, then how is it possible that so many driver genes are found in normal human tissues that do not express cancer? No explanation has been presented on how the SMT can account for, (a) malignant tumors that have no mutations, or (b) normal cells that express driver mutations, but do not develop tumors [14].

(3) The general absence of cancers in chimpanzees despite having about 98% gene and protein sequence identity with humans even at the BRCA1 locus [28,29,30,31]. Despite anatomical differences between the breasts of humans and chimpanzees, breast cancer has never been documented in a female chimpanzee [31]. As DNA replication would be similar in normal tissue stem cells in chimpanzees and humans, the rarity of cancer in all chimpanzee organs undermines the “bad luck” hypothesis of Tomasetti and Vogelstein that cancer risk is due to random mutations arising during DNA replication in normal, noncancerous stem cells [32]. The rarity of cancer in primitive humans and in chimpanzees suggests that environmental factors (diet and lifestyle), rather than genetic mutations, are largely responsible for cancer [31,33]. It is important to remember that nothing in either general biology or in cancer biology makes sense except in the light of evolution [34,35]. 

(4) Theodor Boveri, the person most recognized as the originator of the SMT [36,37], never directly studied cancer and was highly apologetic for his general lack of knowledge about the disease. Indeed, Boveri stated: “*I have no personal experience worth mentioning in any of the numerous specialized fields of tumour research. My knowledge comes almost exclusively from books. Given this, it is inevitable that I am unaware of many reports in the literature, that I overestimate the significance of many known facts and that I do not set enough store by others. But this article will doubtless contain even more serious defects, as is so often the case when an author makes an incursion into a field with which he is unfamiliar*” [38]. Most importantly, Boveri also mentioned that defects in the cytoplasm could just as well be responsible for cancer as defects in the nucleus. The distinguished British geneticist C. D. Darlington also emphasized the importance of the cytoplasm in the origin of cancer [39].

The most compelling evidence against the SMT comes from the nuclear/cytoplasm transfer experiments showing that growth-regulated cells can be produced from tumorigenic nuclei, as long as the tumorigenic nuclei are localized in the cytoplasm containing normal mitochondria [40,41] (Figure 1). Moreover, recent studies show that normal mitochondria can down-regulate multiple oncogenic pathways and abnormal growth in glioma, melanoma, and metastatic breast cancer cells [42,43,44,45,46]. These findings indicate that normal mitochondrial function can suppress dysregulated cell growth regardless of the number of gene or chromosomal abnormalities that might be present in the tumor nucleus. Although the somatic mutations present in the cancer nuclei of developing frogs and mice did not cause dysregulated cell growth, they did abort development suggesting an inhibitory lethal effect on the proliferation of normal cells [40]. If nuclear encoded driver genes were responsible for dysregulated cancer cell growth, then the results from the nuclear/mitochondrial transfer experiments would be opposite to the results shown in Figure 1. When viewed collectively, these findings imply that the nuclear somatic mutations found in many cancers cannot be the primary cause of the disease and seriously challenge the SMT as a credible explanation for the origin of cancer [14]. 

Despite these glaring inconsistencies, the SMT is presented as if it were a settled issue in most current college textbooks of genetics, biochemistry, and cell biology, as well as in the National Cancer Institute in stating that, “*Cancer is a genetic disease—that is, it is caused by changes to genes that control the way our cells function, especially how they grow and divide*” (12 October 2017) [40]. The view of cancer as a genetic disease has become a “silent assumption”, so completely accepted that it is no longer questioned. Could the continued acceptance of the SMT as an explanation for the origin of cancer be based more on *dogmatic ideology* than on rational thought [40,47]? If nuclear somatic mutations cannot be the origin of cancer, then how do cancer cells arise?

## 4. The Mitochondrial Metabolic Theory of Cancer

According to the MMT, cancer arises from a gradual disruption of ATP synthesis through oxidative phosphorylation (OxPhos) leading to compensatory ATP synthesis through substrate level phosphorylation. It is defective OxPhos that ultimately causes most of the genomic changes in cancer, not the reverse. Although Otto Warburg is rightfully credited with the original discovery of cancer as a mitochondrial metabolic disease [50,51], he was unaware of information now available that more strongly supports the linkage between OxPhos deficiency and the origin of cancer (discussed later). The disruption of OxPhos leads to the accumulation of reactive oxygen species (ROS), which are mutagenic and carcinogenic [52,53,54,55,56]. The genomic instability and somatic mutations seen in most cancers arise as a consequence of the chronic production of ROS and acidification of the microenvironment [15,55,57,58,59,60,61]. In other words, the somatic mutations arise as downstream effects rather than as causes of cancer. The information summarized in Figure 1 shows that nuclear genomic mutations alone cannot be the origin of dysregulated cell growth, i.e., the signature phenotype of cancer. Could this information change opinions on the importance of mutations in the origin cancer? It is our view that the MMT can explain better the hallmarks and facts of cancer than can the SMT. 

The MMT is the only theory to provide a credible explanation for the “oncogenic paradox” that has perplexed the cancer field for decades [62,63,64]. Albert Szent-Gyorgyi first described the oncogenic paradox as a specific process (malignant transformation) that could be initiated by a plethora of unspecific events (radiation, asbestos, viral infections, rare inherited mutations, irritation, inflammation, chemicals etc.) [62]. Siddhartha Mukherjee also struggled to understand the paradox in stating on page 285 of his book: “*What beyond abnormal, dysregulated cell division, was the common pathophysiological mechanism underlying cancer?”* [64]. We solved the paradox in showing how the protracted loss of OxPhos, following mitochondrial damage, is the common pathophysiological mechanism responsible for the oncogenic paradox and the origin of cancer (Figure 2).

Since no inherited cancer mutation has been found that is 100% penetrant, inherited cancer mutations are also considered secondary effects and not primary causes of cancer [66,86,87,88]. If an inherited or somatic mutation were to be found in all cancers, it could be considered a primary cause of cancer. Such mutations, however, have not been found. Inherited cancer mutations could cause cancer if they compromise OxPhos function, making OxPhos dysfunction the primary cause of cancer. While mtDNA mutations have been found in many cancers [89], we were unable to find a single pathogenic mutation in the fully sequenced mtDNA of five independently-derived mouse brain tumors [90]. These findings indicate that mtDNA mutations alone cannot be the origin of all cancers. The mtDNA mutations are considered secondary risk factors and can be linked to cancer origin only if they also disrupt OxPhos function [91]. A chronic disruption of ATP synthesis through OxPhos would induce, by necessity, a compensatory energy production through the process of substrate level phosphorylation in both the cytoplasm and in the mitochondria. As normal mitochondrial function maintains cellular differentiation, the rewiring of energy metabolism from respiration to fermentation would cause dedifferentiation and dysregulated proliferation [2,40,50,51,58]. While aerobic fermentation (Warburg effect) is considered another emerging hallmark of cancer [12], the replacement of abnormal mitochondria with normal mitochondria will also reverse this hallmark [44,46,92]. In other words, OxPhos sufficiency will reverse the Warburg effect [44,93]. Hence, the energy transition from respiration to fermentation can explain the major hallmarks of cancer, as described by Hanahan and Weinberg (Figure 2). 

A chronic loss of OxPhos will activate the mitochondrial stress response or the retrograde (RTG) signaling system [75,76,77]. Activation of this system stabilizes Hif-1α and upregulates the expression of c-Myc, key oncogenes necessary for the upregulation of substrate level phosphorylation through the glycolysis and the glutaminolysis pathways, respectively [83]. As plasma membrane pumps are perpetual consumers of ATP, no cell can survive for very long without constant synthesis of ATP for the pumps [94]. The transition from ATP synthesis through OxPhos to ATP synthesis through fermentation thus becomes essential for cell viability. Moreover, the energy transition from OxPhos to fermentation will cause a cell to enter its “default” state. Proliferation is the evolutionary conserved default state of metazoan cells, once freed from active control [3,17]. The mitochondrial OxPhos system provides the active control necessary for maintaining the quiescent or differentiated state. The protracted replacement of ATP synthesis through OxPhos with ATP synthesis through substrate level phosphorylation will cause the cell to enter its default state of proliferation [62]. Szent-Gyorgyi described how unbridled proliferation, driven by fermentation metabolism, was the common phenotype of all cells before oxygen entered the atmosphere some 2.5 billion years ago [62]. Based on the concepts of evolutionary biology, the transition from respiration to fermentation becomes the most logical explanation for the first three hallmarks of cancer involving dysregulated cell growth (Figure 2). 

The acidification of the cancer microenvironment, arising from the excretion of fermentation end products, e.g., lactate and succinate, will initiate angiogenesis. This process, however, can be bi-directional leading to an escalating situation of biological chaos [40,95]. Stabilization of Hif-1α is ultimately responsible for angiogenesis, i.e., the fourth hallmark [82,83,96,97]. As mitochondria control apoptosis [98], evasion of apoptosis would be an expected outcome of dysfunctional mitochondria and can account for the fifth cancer hallmark. While the rewiring of energy production from OxPhos to substrate level phosphorylation can easily explain the first five cancer hallmarks, how might this energy rewiring be linked to metastasis, the sixth major cancer hallmark?

Emerging evidence indicates that metastasis involves transformation of myeloid cells or fusion hybridization between macrophages and transformed epithelial cells [4,99,100,101,102,103,104,105,106,107]. Macrophages and myeloid cells are mesenchymal cells that are already programed to migrate through tissues, to intravasate blood vessels, to function in the circulation, and to extravasate blood vessels for involvement in tissue repair and wound healing [4,108,109,110]. Similar to macrophages, many metastatic cancer cells are immunosuppressive and express phagocytic behavior [100,111,112,113]. The absence of metastasis in crown-gall plant cancers, despite expressing aerobic fermentation (Warburg effect), is due to the absence of a cellular immune system (macrophages and lymphocytes) in plants [4,7]. Macrophages can acquire mitochondria with dysfunctional OxPhos through various fusion hybridization events with neoplastic stem cells in an acidic and hypoxic microenvironment [101,114] (Figure 3). Radiation therapy can also facilitate tumor cell-macrophage/microglial fusion-hybridization thus producing highly invasive metastatic cells, as an unintended consequence [115,116]. It is also interesting that glutamine is a major energy metabolite for cells of the immune system including macrophages [117,118,119]. This fact could account in part for the glutamine dependency of metastatic cancer cells [120,121,122,123]. As macrophages are immunosuppressive, metastatic cells with macrophage properties would be powerful suppressors of the immune system. These properties could contribute to the failure of some immunotherapies [124,125]. The transition from respiration to fermentation can also explain the drug resistance of metastatic tumor cells [126,127]. The drug resistance of tumor cells is due in large part to the replacement of energy synthesis from OxPhos to fermentation [2]. Hence, the control of metastasis can be improved with better knowledge of macrophage biology. 

The macrophage/myeloid origin of metastasis, based on the mitochondrial metabolic theory, should be compared with the epithelial mesenchymal transition (EMT) and mesenchymal epithelial transition (MET) for the origin of metastasis, based on the somatic mutation theory [4,12]. It is unclear how random somatic mutations could be responsible for metastasis, as the metastatic cascade is a non-random phenomenon that is common to many cancer types [5]. Each part of the cascade involves an ordered regulation of evolutionary conserved biological processes. Moreover, the metastatic behavior of cells can occur in the absence of mutations [7,135,136]. The EMT/MET hypothesis has yet to explain how random somatic mutations could transform an epithelial cell into a biologically distinct mesenchymal cell (EMT), and then have these random mutations be suppressed or reversed to allow a transition of the mesenchymal phenotype back to an epithelial phenotype (MET) [12,134]. We consider these fantastical biological transitions as inconsistent with evolutionary biology [85]. In summary, the mitochondrial metabolic theory can explain better the facts of metastasis than can the somatic mutation theory.

## 5. Bidirectional Interactions Involving the Tumor Microenvironment

Bidirectional interactions between the microenvironment and cells in tissues could also alter mitochondrial function and thus the path to neoplasia [57,58]. In contrast to inflammation arising from sepsis and lipopolysaccharides (LPS), which induces acute mitochondrial failure and cell death, the inflammation associated with the origin of cancer is chronic. Chronic inflammation will produce protracted mitochondrial damage [69,137,138,139]. Injury or damage to the mitochondrial electron transport chain can arise from persistent nitric oxide expression in the inflamed microenvironment [139,140]. It is interesting that nitric oxide can inactivate cytochrome c and produce excess ROS without reducing oxygen consumption rates [140]. Andre Nel and colleagues showed how ultrafine particles exacerbate oxidative stress and mitochondrial damage while depleting intracellular glutathione levels in macrophage and epithelial cell lines [69,141]. Nina Bissell’s group together with Bierie and Moses reviewed information showing how chronic inflammation in the microenvironment activates transforming growth factor beta (TGF-β) [69,142,143,144]. Yoon and colleagues showed that TGF-β induces protracted mitochondrial ROS production, which damages respiratory control and enhances senescence in lung epithelial cells [145]. Seoane et al. showed how nuclear genomic instability directly correlated with mitochondrial ROS production [73], while Fosslien described how gradients of TGF-β could alter mitochondrial ATP generation in the morphogenetic field [58,69]. Chronic inflammation, which enhances expression of nitric oxide and TGF-β will damage respiration [69,70]. Most cells suffering respiratory damage will die. According to the MMT, tumors arise only from those cells capable of increasing fermentation in order to compensate for insufficient respiration. Enhanced fermentation prevents cellular senescence [69,146,147]. Although it is clear that inflammation links damaged respiration to the origin of cancer, further studies are necessary to better define how neoplasia can arise through the bidirectional linkage between abnormalities in the microenvironment and respiratory insufficiency within cells. Viewed collectively, these findings indicate that chronic inflammation in the microenvironment is linked to respiratory damage and genomic instability. Additional linkages between respiratory damage, ROS production, and the hallmarks of cancer are described in Figure 2.

Besides the MMT and SMT, the origin of cancer has been described under the tissue organization field theory (TOFT), which also addresses bidirectional interactions in tissue morphogenetic fields [40]. Readers are referred to the excellent work of C. Sonnenschein and A. Soto for a comparative analysis of the TOFT and SMT of carcinogenesis [3,17,148,149,150,151]. According to these investigators any conclusion regarding data interpretation is valid under the SMT because no alternative concept is ever disproved and abandoned. The lack of data fit with the SMT is attributed to the unfathomable complexity of biology. Put simply, something can be anything and its opposite when viewed under the SMT [17]. 

## 6. Mitochondrial Substrate Level Phosphorylation in Cancer

A foundational principle in biological evolution is that structure determines function. Abnormalities in structure will cause abnormalities in function. As mitochondrial structure is intimately connected to OxPhos, abnormalities in the structural integrity of mitochondria and in mitochondrial associated membranes (MAM) will compromise OxPhos function [83,152,153,154,155,156,157,158,159,160]. We recently reviewed evidence showing that abnormalities in mitochondria number, structure, and function have been documented in most major human cancers including those from bladder, mammary, colorectal, nervous system, kidney, blood (lymphomas), liver, lung, skin, bone, ovary, pancreas, prostate, retina, and salivary glands, among others [83]. These abnormalities, together with evidence that cancer cells can grow and survive in cyanide and in hypoxia (0.1% oxygen) indicate that OxPhos “cannot” be responsible for sufficient ATP synthesis in most cancer cells [83,161,162,163,164,165,166,167]. Although aerobic glycolysis is linked to malignancy in most tumors [168,169], the amount of ATP synthesized through glycolysis can be ambiguous in cancer cells. We reviewed information showing that many malignant cancers express the pyruvate kinase 2 isoform (PKM2), which produces pyruvate with little ATP synthesis in the last step of glycolysis [83,84,170]. The high glucose consumption in cancer cells is used more for the synthesis of growth metabolites through glycolysis and the pentose pathway than for the synthesis of ATP [84,170,171,172]. Glucose carbons are major contributors to the pentose phosphate pathway and to serine synthesis through one carbon metabolism [171,173,174,175]. We also described how the accumulation of lipid droplets in cancer cells is the manifestation of defective OxPhos [83]. Although fatty acids cannot be used as a fuel in tumor cells with defective OxPhos, they can enhance glucose and glutamine fermentation through a range of mechanisms [83,156,176,177,178,179]. Furthermore, we described how measurements of oxygen consumption rates (OCR) in cultured cancer cells can be misinterpreted as ATP synthesis through OxPhos, i.e., OCR does not necessarily indicate ATP synthesis through OxPhos [83,140,180,181]. It should be recognized that ATP synthesis is the common requirement for tumor cell growth and survival regardless of the cellular or the genetic heterogeneity found within tumors [83]. If OxPhos and glycolysis are not the origin of most ATP synthesis in tumors, then where would cancer cells obtain the energy necessary for their survival and growth? 

We described how mitochondrial substrate level phosphorylation (mSLP) in the glutamine-driven glutaminolysis pathway can compensate for reduced ATP synthesis through both OxPhos and glycolysis [83,182]. The succinate-CoA ligase reaction in the TCA cycle can produce sufficient ATP for cancer cell growth and viability while maintaining the adenine nucleotide translocase in forward mode, thus preventing the reverse-operating F_0_-F_1_ ATP synthase from depleting cytosolic ATP reserves [83,182,183]. It is important to emphasize that dissipation of the protonmotive force and reversal of the F_0_-F_1_ ATP synthase will occur following an OxPhos reduction of ~50% [184,185]. The F_0_-F_1_ ATP synthase operates in forward mode, i.e., generating ATP, only when mitochondria are sufficiently polarized. It would not be possible for the F_0_-F_1_ ATP synthase to generate ATP under a loss of ETC operation on the order of 45–50% [93,182]. Such a situation would cause ATP hydrolysis thus pumping protons out of the matrix. It is the reversal of the ATP synthase that gives mitochondrial substrate level phosphorylation a critical role in providing ATP within the matrix under OxPhos deficiency.

The glutaminolysis pathway can produce high-energy phosphates through the sequential metabolism of *glutamine* -> *glutamate* -> *alpha-ketoglutarate* -> *succinyl CoA* -> *succinate* (Figure 4). Glutamine is the only amino acid that can generate significant ATP synthesis through mSLP in the glutaminolysis pathway [182]. With the exception of glutamate and glutamine, the catabolism of most other amino acids would *expend* high-energy phosphates during metabolic inter-conversions before becoming succinyl-CoA and cannot therefore effectively replace glutamine for ATP synthesis. Glutamine is the obligate nitrogen donor in at least three independent steps for purine synthesis including the phosphoribosylpyrophosphate amidotranferase, the phosphoribosylformylglycinamidine synthetase, and the GMP synthetase [182]. Glutamine is also important in two independent enzymatic steps for pyrimidine synthesis, i.e., the carbamoyl phosphate synthetase II step and CTP synthetase step [182,186,187]. Glutamine-derived glutamate is also the primary nitrogen donor for the synthesis of non-essential amino acids including asparagine [186]. This is interesting because asparagine has been considered a growth metabolite for GBM and other tumors [182,188,189]. The glutamine: fructose-6-phosphate amidotransferase reaction transfers the amide nitrogen of glutamine to form glucosamine-6-phosphate, a precursor for N-linked and O-linked glycosylation that is needed for hexosamine synthesis [182,190]. Although some have suggested that glutamine can be metabolized to lactate through the malic enzyme to produce NADPH for lipid biosynthesis [78,191], most other findings, however, indicate that little glutamine is metabolized to lactate in cancer cells [162,182,192,193,194]. We consider succinate, not lactate, as the end product of the glutaminolysis pathway [83,182].

As *Q* is the letter designation for glutamine, we have described glutamine-driven ATP synthesis in cancer cells as the Q effect to distinguish it from that involving the aerobic fermentation of glucose, i.e., the Warburg effect [83,182]. Both the Warburg effect and the Q effect are downstream effects of compromised OxPhos function (Figure 2). Hall et al., also showed that aerobic glycolysis becomes elevated in proportion to the *V600EBRAF*-induced OxPhos dysfunction in melanoma cells thus linking the Warburg effect directly to OxPhos dysfunction in these cells [93]. In other words, aerobic glycolysis is an effect and not the cause of OxPhos dysfunction ruling out teleological arguments for a *purpose* of glycolysis in cancer [195,196]. Unfortunately, the role of glutaminolysis and mSLP in cellular energy metabolism was unknown to Warburg, as this information was discovered only after, or towards the end of his career [83,197,198,199,200]. Warburg did not envision amino acid (glutamine) fermentation as a second major compensatory energy source to OxPhos in his theory of cancer. We consider glutamine-driven mSLP as a primary mechanism for ATP synthesis in tumor cells that express ultrastructural abnormalities in mitochondrial cristae, that overexpress PKM2, and that can grow in cyanide or in hypoxic (0.1% oxygen) environments.

mSLP is the metabolic hallmark of tumor cell proliferation whether growth is either in vivo or in vitro [83]. Indeed, Chen et al. showed that ATP synthesis through mSLP could compensate for ATP syntheses deficiencies in either glycolysis or OxPhos [201]. mSLP has been documented as a source of ATP synthesis in cancer cells [120,202,203,204]. The Crabtree effect can induce aerobic lactate fermentation in non-tumorigenic cells that proliferate in vitro but does not occur in normal cells that proliferate in vivo [83,205]. The activation of immune cells in vitro can induce aerobic fermentation (Warburg effect) as an artifact of the abnormal environment [206,207]. It is important to recognize that aerobic fermentation does not occur in proliferating non-transformed cells grown in vivo, for example, in regenerating liver cells, in normal colon cells, or in proliferating T-Cells and B-Cells [207]. Non-fermentable fatty acids and butyrate are used as respiratory fuels for liver and colon regeneration, whereas glucose respiration drives proliferation of T-Cells and B-Cells [182,207,208,209,210]. Warburg also described how aerobic fermentation could confuse the issue of cancer cell metabolism and should not be considered as a test for cancer cells [51]. Support for Warburg’s position that aerobic glycolysis (fermentation) confuses the issue of cancer metabolism came from the studies of R. J. O’Connor who misinterpreted the connection of oxygen consumption to cell division in the early chick embryo [50,51,83,211]. It should also be recognized that “anaerobic” fermentation, not “aerobic” fermentation, is largely responsible for cell division in the early embryo [50,51]. Confusion over the association of OxPhos with oxygen consumption rates and a general failure to recognize the role of mSLP as a compensatory energy mechanism can explain in large part how some investigators might consider OxPhos as functional and responsible for ATP synthesis in tumor cells [83]. Hence, recognition of mSLP, as the missing link in the MMT of cancer, will refocus the cancer field on mitochondrial OxPhos dysfunction as the origin of cancer.

## 7. Cancer Management Based on the MMT

If a capability is necessary for tumor growth, then the inhibition of this capability should be essential for an effective management of cancer [1,12]. The necessary capability for cancer cells, based on the MMT, is the fermentation metabolism needed for the synthesis of growth metabolites and ATP through the glycolytic and glutaminolysis pathways [83]. In contrast to a plethora of opinions on the origin and behavior of cancer cells [212], we view cancer as a relatively simple disease dependent almost exclusively on the availability of glucose and glutamine for survival. In short, no cancer cell can survive for very long without growth metabolites and ATP. We recently showed how the simultaneous targeting of the glycolysis and the glutaminolysis pathways can significantly improve progression free and overall survival in orthotopic syngeneic GBM mouse models and in a long-term survival patient with an *IDH1(R132H)*-mutant GBM [123,213]. We consider the *IDH1(R132H)* as a “therapeutic mutation” due to its ability to improve progression free and overall survival in glioblastoma patients [213]. We described how the *IDH1(R132H)* mutation can act synergistically with ketogenic metabolic therapy (KMT) to simultaneous target both the glycolysis and glutaminolysis pathways (Figure 4). It is possible that additional somatic mutations might be found that also have therapeutic potential in targeting the glycolysis and glutaminolysis pathways. In contrast to unacceptably toxic *Rube Goldberg*-type therapeutic strategies based on the SMT [124,125,214], the simplest and most parsimonious non-toxic strategy is to simply restrict availability of the two fuels that are necessary and sufficient for cancer cell growth and survival, i.e., glucose and glutamine [1,83,123,213]. 

The simultaneous restriction of glucose and glutamine for the metabolic management of cancer can be best achieved using a “press-pulse” therapeutic strategy [1]. This management strategy was derived from the concepts of paleobiology and evolutionary adaptation [218,219,220]. N. Arens and I. West described how the simultaneous occurrence of “press-pulse” disturbances was considered the mechanism responsible for the extinction of organic populations during prior evolutionary epochs [218]. In adapting this concept to cancer management, “press” disturbances would eliminate the weakest cancer cells, while growth-restricting the heartiest cancer cells. In contrast, a “pulse” disturbance is an acute treatment that would kill most, but not all cancer cells. It is only when both the press and the pulse disturbances are used simultaneously that mass extinction of all cancer cells becomes possible [1]. As a prototype example, we used ketogenic metabolic therapy (KMT) as a press and the pan glutaminase inhibitor DON (6-diazo-5-oxo-L-norleucine) as a pulse to manage the growth of preclinical glioblastoma [123]. KMT restricts glucose availability while elevating ketone bodies and thus induces a competition between normal cells and tumor cells for glucose [221]. Ketone bodies and fatty acids are non-fermentable and cannot replace glucose in cells with defective mitochondria. Ketone body elevation under fasting (nutritional ketosis) can allow blood glucose to reach extremely low levels (0.5 mM or 9 mg/dL without adverse effects [222]. Ketone bodies also suppress glucose consumption in the brain, i.e., the largest consumer of glucose in the body [223,224]. Additionally, KMT will reduce inflammation in the tumor microenvironment thus restricting tumor cell invasion [1,115,225,226]. Glucose restriction will inhibit tumor cell growth, as the glucose carbons are needed for the synthesis of growth metabolites through the pentose phosphate and glycolysis pathways. DON will restrict availability of the glutamine nitrogen and the glutamine carbons that are necessary for the synthesis of nitrogen-containing metabolites and the synthesis ATP through mSLP in the glutaminolysis pathway. Hence, the simultaneous restriction of glucose and glutamine, while under KMT, will reduce acidification in the tumor microenvironment and will target both the glycolysis and glutaminolysis pathways that are essential for tumor cell growth and survival. 

The success in dealing with environmental stress and disease is dependent on the integrated action of all cells in the organism according to R. Potts of the Smithsonian Institution of Human Origins [219,220]. This integrated action depends on the flexibility of each cell’s genome, which responds to both internal and external signals according to the needs of the organism. Adaptability to abrupt environmental change is a property of the genome, which was selected for in order to ensure survival under environmental extremes [35,227]. More specifically, only those cells possessing flexibility in nutrient utilization will be able to survive under nutrient stress. Environmental forcing has selected for genomes that are most capable of adapting to change in order to maintain metabolic homeostasis [10,219,220,227]. The genomes of most cancer cells, however, contain numerous types of pathological mutations, chromosomal rearrangements, and aneuploidy. These genomic defects, together with mitochondrial dysfunction, will prevent the flexibility needed for rapid adaption to nutrient stress. Simply stated, the genomic defects in cancer cells will prevent the adaptive versality needed to survive physiological and nutrient stress. Hence, press-pulse therapeutic strategies should be highly effective in providing long-term management and possible resolution of most cancers [1].

According to the SMT, tumor cells have a growth advantage and are more fit than normal cells. This view is not only inconsistent with Darwin’s theory of evolution but is also inconsistent with Potts’ theory of adaptive versatility [1,35,219,227]. Dysregulated cell growth should not be considered an advantage over regulated cell growth, as the regulated growth rate of regenerating liver cells is faster than that of most tumor cells [50,228,229]. Moreover, the rapid growth of regenerating liver cells is dependent on fatty acid-driven OxPhos rather than on glucose-driven fermentation [209,210]. In contrast to cancer cells, little lactate is produced in regenerating liver cells. Indeed, glucose inhibits the growth of regenerating liver cells [209,229]. These findings in regenerating liver, as well as those in proliferating cells from the gut and immune system (mentioned above), do not support suggestions that aerobic glycolysis is common to all types of proliferating cells. It is clear that fermentation-dependent tumor cells do not have a growth advantage over OxPhos-dependent regenerating liver cells. 

The resistance of tumor cells to drug treatments can make them appear more fit than normal cells. Drug resistance is due in large part to the fermentation metabolism that drives dysregulated growth thus facilitating cancer cell survival in hypoxic environments [126,127]. As long as tumor cells have access to glucose and glutamine they can appear as more fit than normal cells. A whole-body transition from glucose-driven metabolism to ketone body-driven metabolism will produce a powerful press disturbance on any tumor cell dependent on glycolysis for growth. Moreover, ketone body metabolism enhances the ΔG’ATP hydrolysis in normal cells from −56 kJ/mole to −59 kJ/mole, thus providing normal cells with an energetic advantage over tumor cells, which are limited to energy generation through fermentation [213,227,230,231]. Based on the tenets of evolutionary biology and on the concepts of the MMT, tumor cells are *not* more fit and do *not* have a growth advantage over normal cells. It is important to mention again that nothing in cancer biology makes sense except in the light of evolution [35]. 

Finally, it is interesting to reflect on the closing comments from Warburg’s seminal 1956 paper [50]. *“If the explanation of a vital process is its reduction to physics and chemistry, there is today no other explanation for the origin of cancer cells, either special or general. From this point of view, mutation and carcinogenic agent are not alternatives, but empty words, unless metabolically specified. Even more harmful in the struggle against cancer can be the continual discovery of miscellaneous cancer agents and cancer viruses, which, by obscuring the underlying phenomena, may hinder necessary preventive measures and thereby become responsible for cancer cases.”* The underlying phenomenon in our view is the near exclusive dependence on the fermentation metabolism of glucose and glutamine for the growth and the survival of most, if not all cancers. Cancer management and prevention will be improved significantly once these concepts become more widely recognized and accepted.

## 8. Conclusions

Information is reviewed showing that the MMT can explain better the facts of cancer than can the SMT. Most tumor cells, regardless of their tissue origin or genomic abnormalities, are largely dependent on fermentation metabolism through the glycolysis and the glutaminolysis pathways for the synthesis of growth metabolites and ATP. No tumor cell can grow or survive without metabolites or energy. The simultaneous targeting of these pathways offers a non-toxic therapeutic strategy for effectively managing most cancers. The simplest and most parsimonious strategy for managing cancer under the MMT is to restrict availability of glucose and glutamine while placing the whole body in a state of nutritional ketosis. 

## Figures and Tables

**Figure 1 metabolites-11-00572-f001:**
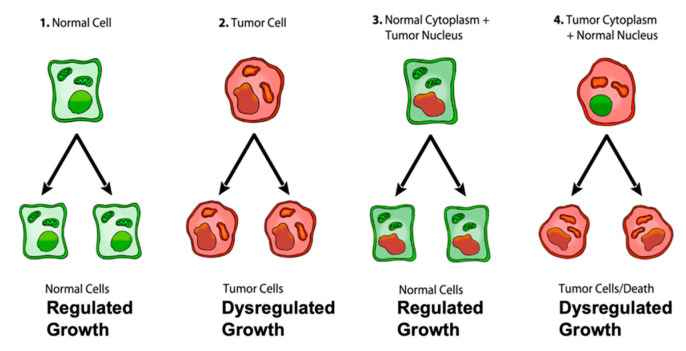
Role of the nucleus and mitochondria in the origin of tumors. Normal cells are shown in green with nuclear and mitochondrial morphology indicative of normal gene expression and OxPhos function, respectively. Tumor cells are shown in red with abnormal nuclear and mitochondrial morphology indicative of genomic instability and abnormal OxPhos function, respectively. *“(1) Normal cells beget normal cells with regulated growth. (2) Tumor cells beget tumor cells with dysregulated growth. (3) Transfer of a tumor cell nucleus into a normal cytoplasm begets normal cells that have regulated growth, despite the presence of the tumor-associated genomic abnormalities. (4) Transfer of a normal cell nucleus into a tumor cell cytoplasm begets dead cells or tumor cells with dysregulated growth”.* The general reproducibility of the findings across a broad range of tumor types, animal species, and experimental techniques is notable in light of major concerns regarding the irreproducibility of scientific results published in prestigious journals [40,48,49]. *“The results of these experiments are profound in showing that nuclear genomic defects alone cannot account for the origin of tumors and that normal-functioning mitochondria can suppress tumorigenesis”.* Original diagram from Jeffrey Ling and Thomas N. Seyfried with permission [41].

**Figure 2 metabolites-11-00572-f002:**
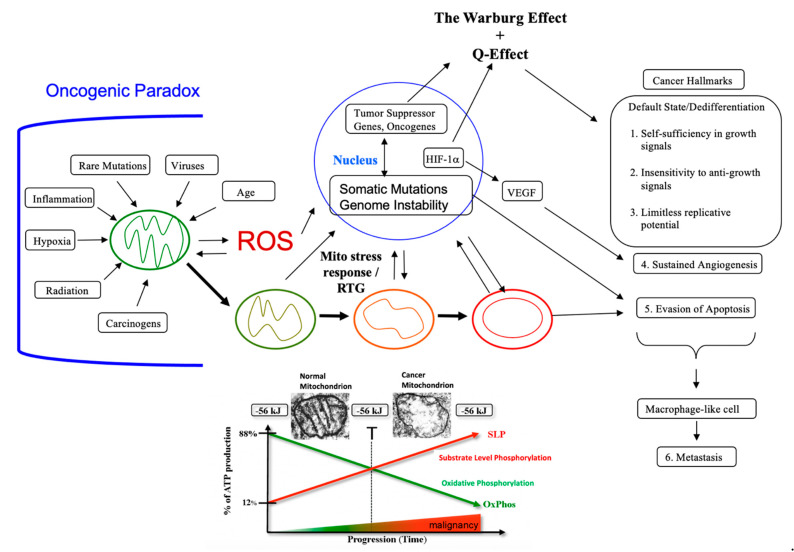
Cancer as a Mitochondrial Metabolic Disease. Cancer can arise from any number of unspecific influences (risk factors) that would alter the number, structure, and function of mitochondria thus affecting energy production through OxPhos. Unspecific cancer risk factors can include, age, viral infections, the *Ras* oncogene, rare inherited mutations, chronic inflammation, intermittent hypoxia, radiation exposure, chemical carcinogens etc. [2,65,66,67,68]. Any of these risk factors could cause chronic damage to OxPhos thus increasing the production of reactive oxygen species (ROS), which would ultimately link to the six major hallmarks of cancer [2,12,68]. The process by which each of these unspecific risk factors can chronically damage OxPhos was described previously in detail [16,66,69,70]. Excessive ROS, mostly generated from OxPhos dysfunction, are carcinogenic and mutagenic and would cause significant damage to lipids, proteins, and nucleic acids in both the mitochondria and the in the nucleus [71]. Nuclear genomic instability, including the vast array of somatic mutations and aneuploidy, would arise because of ROS damage together with extracellular acidification and inflammation through a bidirectional interaction between the provocative agent and cells within a tissue [1,2,57,72,73]. Indeed, mutations in the *p53* tumor suppressor gene and genomic instability have been linked directly to OxPhos deficiency and mitochondrial ROS production in cancer stem cells [55,74]. Fermentation metabolism and ROS formation underlie the hyperproliferation of tumor cells. A gradual reduction in OxPhos efficiency would elicit a mitochondrial stress response through retrograde (RTG) signaling [69,75,76,77]. RTG activation would cause persistent expression of various oncogenes, e.g., *Hif-1a* and c-*Myc*, that upregulate receptors and enzymes in both the glycolysis and the glutaminolysis pathways [75,78,79,80,81,82]. Oncogenes become facilitators of fermentation metabolism. ATP synthesis through mSLP (Q effect) will compensate for lost ATP synthesis through OxPhos or from PKM2 expression in glycolysis [83,84]. The path to carcinogenesis will occur only in those cells capable of sustaining energy production through substrate level phosphorylation, (SLP). Cells unable to replace OxPhos with SLP, e.g., CNS neurons or cardiomyocytes, would die and rarely become tumorigenic. Despite the shift from respiration to SLP, the ΔG’ATP hydrolysis remains fairly constant at approximately −56 kJ, indicating that the energy from SLP compensates for the reduced energy from OxPhos. When respiration becomes unable to maintain energy homeostasis, the RTG will initiate oncogene up-regulation and tumor suppressor gene inactivation. Protracted RTG activation becomes necessary to maintain the viability of incipient cancer cells. Genomic instability will arise as a secondary consequence of protracted mitochondrial stress from disturbances in the intracellular and extracellular environments. Metastasis arises from respiratory damage in cells of myeloid/macrophage origin either directly or after fusion hybridization with epithelial-derived tumor cells [4,85]. Tumor progression and degree of malignancy is linked directly to ultrastructure abnormalities (mitochondrial cristolysis) and to the energy transition from OxPhos to substrate level phosphorylation (Warburg effect and Q effect) [83]. The **T** signifies an arbitrary threshold when the shift from OxPhos to SLP becomes irreversible. This scenario links all major cancer hallmarks to an extrachromosomal and epigenetic respiratory dysfunction and can explain the oncogenic paradox [70]. Reprinted with modifications from [68,83].

**Figure 3 metabolites-11-00572-f003:**
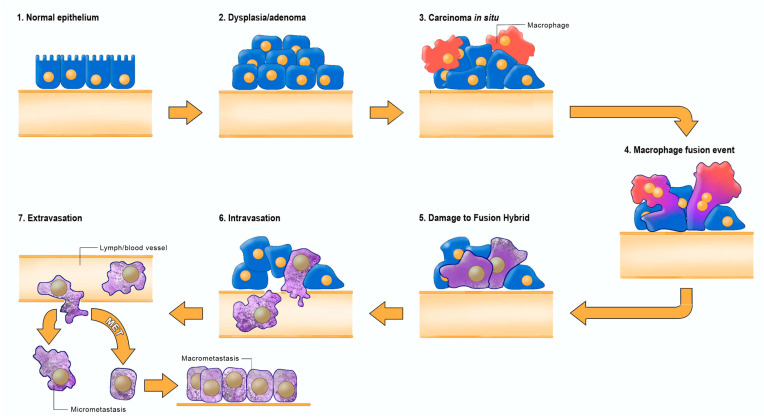
Fusion-hybrid hypothesis for cancer cell metastasis. According to the fusion hybrid hypothesis, metastatic cancer cells can arise following fusion-hybridization between neoplastic epithelial cells and myeloid cells (macrophages). The fusion hybrid hypothesis originated with the work of Aichel in 1911 and was expanded by the Pawelek and the Munzarova groups [105,128,129,130,131,132]. Macrophages are known to invade in situ carcinoma as if it were an unhealed wound [95,109,133]. This creates a protracted inflammatory microenvironment leading to fusion hybridization between the neoplastic epithelial cell and the mesenchymal macrophage. Mitochondrial damage becomes the driver for the neoplastic transformation of the epithelial cell and of the fusion hybrids. Inflammation damages mitochondria leading to enhanced fermentation and acidification of the microenvironment. The gradual replacement of normal macrophage mitochondria with dysfunctional mitochondria in the hybrid cell cytoplasm leads to rogue behavior in cells that naturally possess the capability to, (1) move through tissues, (2) suppress the immune system, (3) enter (intravasate), and to exit (extravasate) the circulation. In addition to explaining the “seed-soil” hypothesis of metastasis, the fusion hybrid hypothesis can also explain how metastatic cells can re-capitulate the epithelial characteristics of the primary tumor at secondary micro-metastatic growth sites [4,85]. Furthermore, this hypothesis can explain the phenomenon of mesenchymal epithelial transition without invoking a mutation suppression mechanism. See text for more details. Modified from [85,134].

**Figure 4 metabolites-11-00572-f004:**
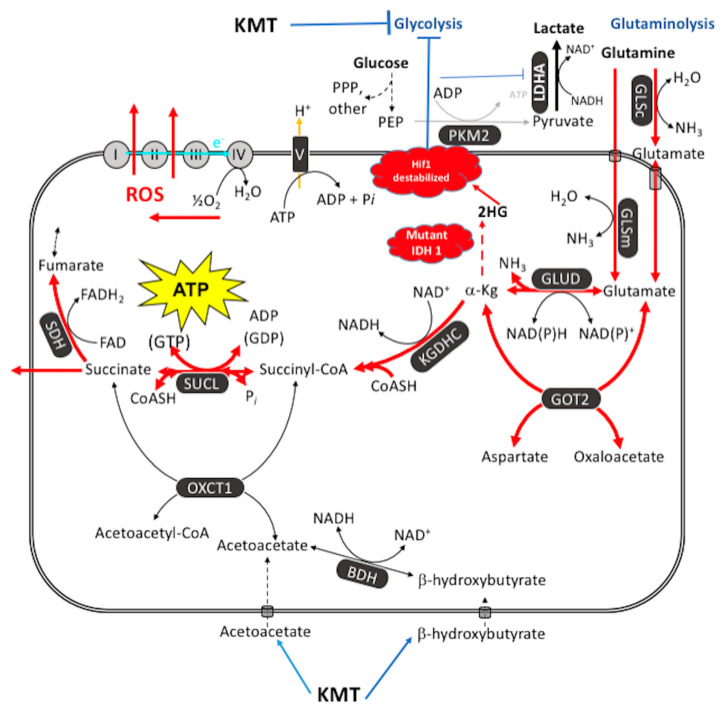
Glutamine-driven mSLP as a major source of ATP synthesis in cancer cells and inhibition by the therapeutic *IDH1(R132H)* mutation. ATP synthesis through mSLP at the succinate-CoA ligase reaction (SUCL) in the glutaminolysis pathway (red) can compensate for inefficient ATP synthesis through OxPhos in tumor cells with mitochondrial abnormalities [83]. mSLP can also compensate for inefficient ATP synthesis through glycolysis in cancer cells that express the cytoplasmic PKM2 isoform, which produces less ATP than the PKM1 isoform [84]. These bioenergetic compensations will hold the ΔG’_ATP_ hydrolysis at −56 kJ/mole, thus maintaining cancer cell viability in either the presence or the absence or oxygen. Oxygen consumption in cancer cells is used more for production of ROS, which are carcinogenic and mutagenic, than for ATP synthesis. Moreover, mSLP will maintain the forward operation of the adenine nucleotide translocase thus preventing depletion of cytosolic ATP reserves from the reverse operation of the F0-F1 ATP synthase [84,182]. Release of succinate to the cytoplasm can help stabilize Hif-1α, thus linking lactic acid fermentation through glycolysis to glutamine fermentation through glutaminolysis. Ketogenic metabolic therapy (KMT) will reduce availability of glucose to the glycolytic and the PPP pathways while diverting CoA from succinate to acetoacetate under the metabolism of ketone bodies (β-hydroxybutyrate and acetoacetate) thus indirectly reducing ATP synthesis through the SUCL reaction. The therapeutic *IDH1* mutation will further reduce ATP synthesis through mSLP by increasing synthesis of 2-hydroxyglutarate (2HG) from α-ketoglutarate and thus reduce the succinyl CoA substrate for the SUCL reaction [83,215]. Besides its potential effect in reducing glutaminolysis, 2-hydroxyglutarate can also target multiple HIF1α-responsive genes and enzymes in the glycolysis pathway thus limiting synthesis of metabolites and one-carbon metabolism needed for rapid tumor growth [83,182,216,217]. The down regulation of Hif1-α-regulated lactate dehydrogenase A (LDHA), through the action of both KMT and the *IDH1* mutation, would reduce extracellular lactate levels thus reducing microenvironment inflammation and tumor cell invasion. The simultaneous restriction of glucose and glutamine, while elevating circulating ketone bodies, will stress the majority of signaling pathways necessary for maintaining tumor cell viability [123,182]. See text for additional details. KMT = ketogenic metabolic therapy; 2HG = 2-hydroxyglutarate; BDH  =  β-hydroxybutyrate dehydrogenase; FAD  =  flavin adenine dinucleotide; GLSc  =  glutaminase, cytosolic; GLSm  =  glutaminase, mitochondrial; GLUD  =  glutamate dehydrogenase; GOT2  =  aspartate aminotransferase; KGDHC  =  α-ketoglutarate dehydrogenase complex; LDH: lactate dehydrogenase; NME  =  nucleoside diphosphate kinase; OXCT1  =  succinyl-CoA:3-ketoacid coenzyme A transferase 1; PC  =  pyruvate carboxylase; PDH  =  pyruvate dehydrogenase; PEP  =  phosphoenolpyruvate; PKM2  =  pyruvate kinase M2; SDH: succinate dehydrogenase; SUCL  =  succinate-CoA ligase. Reprinted with modifications from [83,213].

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
