# Peer review of "Can the Mitochondrial Metabolic Theory Explain Better the Origin and Management of Cancer than Can the Somatic Mutation Theory?"

_metabolites, 2021, doi:10.3390/metabo11090572_

Round 1

Reviewer 1 Report

In this concept article, the authors argued that the historical theory, so-called "somatic mutation theory" (SMT), failed to capture cancer's sophisticated and multifaceted dysregulations. Throughout the article, "mitochondrial metabolic theory" (MMT) is demonstrated to describe the origins and progression of cancer better.

-Dysregulations of mitochondria, either from mtDNA or from its biochemical processes, need to be accumulated before having an impact on the cell. The process is heavily influenced by cellular dysfunction and the altered microenvironment. It is a bidirectional interaction that eventually leads to the disease phenotype. Could it be discussed further to demonstrate the merit of the proposed theory?

-It goes without saying that dysregulations of mitochondria play an important role in cancer survival, progression, and metastasis. It is well described throughout the manuscript. However, the causal relationship, i.e., the causes of all those effects, should be demonstrated clearly to indicate the merit of the MMT. My concerns are: (1) the SMT and its related components are now not accepted as a universal system to explain the origin and development of cancer. I wonder if it is worth arguing about a replacement? (2) How much could the microenvironment contribute to the genomic instability, and how could the microenvironment change in the first place? (3) I would need more evidence to be convinced that the somatic mutations are the "effects," how about "partial relationship"? (4)

-With the accumulating evidence, we could generally establish few major hallmarks of cancer. These hallmarks are, however, highly convoluted via complex interactions at different molecular levels. In my humble opinion, it suggests that more quantitative metrics are required to find the temporal and spatial events of what happened in the carcinogenesis and the later processes. Perhaps, with quantitative information, we could establish a model to capture and "weights" individual events and their impact from one to another. We could then elucidate the potential targets for proper interventions.

-I would suggest the true value of this article is about an in-depth discussion on the hallmarks of cancer with an emphasis on the role of mitochondria and metabolic aberrations.   

-I wonder if Figure 2 was reprinted with necessary permission from the original article? Please kindly check with the authors/publisher to avoid unnecessary conflicts.

Author Response

Reviewer Comments, MMT vs SMT for Metabolites

We thank the reviewer for the insightful comments and suggestions to improve our concept article. We have provided a point-by-point response to all comments and have underlined all changes in the text of our revised manuscript.

Reviewer #1

  1. In this concept article, the authors argued that the historical theory, so-called "somatic mutation theory" (SMT), failed to capture cancer's sophisticated and multifaceted dysregulations. Throughout the article, "mitochondrial metabolic theory" (MMT) is demonstrated to describe the origins and progression of cancer better.

-Dysregulations of mitochondria, either from mtDNA or from its biochemical processes, need to be accumulated before having an impact on the cell. The process is heavily influenced by cellular dysfunction and the altered microenvironment. It is a bidirectional interaction that eventually leads to the disease phenotype. Could it be discussed further to demonstrate the merit of the proposed theory?

Response: We thank the reviewer for this question.  Indeed, a bidirectional interaction between the microenvironment and cells in tissues could alter mitochondrial function and thus the path to neoplasia.  We added a new section to the revised manuscript on pages 12-13 that addresses bidirectional interactions.

  1. -It goes without saying that dysregulations of mitochondria play an important role in cancer survival, progression, and metastasis. It is well described throughout the manuscript. However, the causal relationship, i.e., the causes of all those effects, should be demonstrated clearly to indicate the merit of the MMT. My concerns are:

(a) the SMT and its related components are now not accepted as a universal system to explain the origin and development of cancer. I wonder if it is worth arguing about a replacement?

Response: Unfortunately, the majority of workers in the cancer field today accept the SMT as the universal system to explain the origin and development of cancer.  This is why it becomes important to review the evidence showing that the MMT can explain better the origin and development of cancer than can the SMT. 

(b) How much could the microenvironment contribute to the genomic instability, and how could the microenvironment change in the first place?

Response: We addressed this point above and have now added this information in the revised manuscript on pages 12-13. 

(c) I would need more evidence to be convinced that the somatic mutations are the "effects," how about "partial relationship"?

Response: The nuclear/mitochondrial transfer experiments provide the strongest evidence against the SMT.  Normal mitochondria suppress dysregulated cell growth regardless of the number and types of somatic mutations present in the nucleus (doi: 10.3389/fcell.2015.00043).  These findings indicate that somatic mutations alone cannot be the origin of cancer.  It is interesting, however, that somatic mutations abort development, but do not cause dysregulated cell growth, the signature feature of cancer.  ROS, which are carcinogenic and mutagenic in excessive amounts, are the product of damaged respiration and contribute significantly to the origin of the random somatic mutations seen in cancers.  As no inherited cancer mutation has been found that is 100% penetrant, inherited cancer mutations are also considered secondary effects and not primary causes of cancer.  If an inherited or somatic mutation were to be found in all cancers, it could be considered a primary cause of cancer.  Such mutations have not been found. Moreover, new information shows that normal human tissues will collect large numbers of somatic “driver mutations” without causing dysregulated cell growth.  Viewed together these findings, which we review in the manuscript, indicate that somatic mutations are downstream effects of chronic respiratory dysfunction.  Inherited cancer mutations can cause cancer only if they compromise OxPhos function, making OxPhos dysfunction the primary cause of cancer. 

Is a partial relationship possible?  Response: A partial relationship could be possible through the following sequence of events. A, Chronic OxPhos damage leads to stabilization of Hif-1a through retrograde signaling.  B, Hif-1a stabilization increases lactic acid fermentation, which would increase extracellular acidification. C, Extracellular acidification contributes to higher levels of ROS thus increasing genomic damage.  The references supporting these statements are included in the revised manuscript.

  1. -With the accumulating evidence, we could generally establish few major hallmarks of cancer. These hallmarks are, however, highly convoluted via complex interactions at different molecular levels. In my humble opinion, it suggests that more quantitative metrics are required to find the temporal and spatial events of what happened in the carcinogenesis and the later processes. Perhaps, with quantitative information, we could establish a model to capture and "weights" individual events and their impact from one to another. We could then elucidate the potential targets for proper interventions.

Response: We agree with the reviewer that a model, which could capture the "weights" of individual events and their impact from one to another, would be important for a better understanding of cancer.  Hopefully the information in our concept paper will provide a framework for establishing the quantitative metrics needed for better defining the temporal and spatial events in carcinogenesis and the later processes.

  1. -I would suggest the true value of this article is about an in-depth discussion on the hallmarks of cancer with an emphasis on the role of mitochondria and metabolic aberrations. 

Response: Yes, we agree and thank you for this comment.

  1. -I wonder if Figure 2 was reprinted with necessary permission from the original article? Please kindly check with the authors/publisher to avoid unnecessary conflicts.

Response: Figure 2 was created by TNS and first appeared in Nutrition and Metabolism; an open access journal (http://www.nutritionandmetabolism.com/content/7/1/7).  The journal is part of creative commons which permits unrestricted use, distribution, and reproduction in any medium, provided the original work is properly cited. The figure is properly cited.

Reviewer 2 Report

This concept paper represents a thoroughly argued viewpoint on the central role of reprogrammed energy metabolism in cancer. It points to the importance of ATP syntheis, in cancer cells,  through glutamine-dependent mitochondrial substrate level phosphorylation. Understanding of metabolic reprogramming in cancer, that underlyes transition of cancer cells between different phenotypic states regarding differentiation and mitotic capability, is of utmost importance. Already  Warburg, despite the lack of many information available to us now, envisioned in his seminal paper "On the origin of cancer cells":
"If the explanation of a vital process is its reduction to physics and chemistry, there is today no other explanation for the origin of cancer cells, either special or general. From this point of view, mutation and carcinogenic agent are not alternatives, but empty words, unless metabolically specified. Even more harmful in the struggle against cancer can be the continual discovery of miscellaneous cancer agents and cancer viruses, which, by obscuring the underlying phenomena, may hinder necessary preventive measures and thereby become responsible for cancer cases."
This manuscript is well writen, and may be of interest to a broad readership, as it  comprehensively summarizes important aspects and critically compares the capacity of somatic mutation theory and mitochondrial metabolic theory to explain the origin of cancer. 

Author Response

We thank the reviewer for the insightful suggestion to improve our concept article. We have provided a point-by-point response to all comments and have underlined all changes in the text of our revised manuscript.

Reviewer #2

This concept paper represents a thoroughly argued viewpoint on the central role of reprogrammed energy metabolism in cancer. It points to the importance of ATP synthesis, in cancer cells, through glutamine-dependent mitochondrial substrate level phosphorylation. Understanding of metabolic reprogramming in cancer, that underlies transition of cancer cells between different phenotypic states regarding differentiation and mitotic capability, is of utmost importance. Already Warburg, despite the lack of many information available to us now, envisioned in his seminal paper "On the origin of cancer cells ":

"If the explanation of a vital process is its reduction to physics and chemistry, there is today no other explanation for the origin of cancer cells, either special or general. From this point of view, mutation and carcinogenic agent are not alternatives, but empty words, unless metabolically specified. Even more harmful in the struggle against cancer can be the continual discovery of miscellaneous cancer agents and cancer viruses, which, by obscuring the underlying phenomena, may hinder necessary preventive measures and thereby become responsible for cancer cases."

This manuscript is well written, and may be of interest to a broad readership, as it comprehensively summarizes important aspects and critically compares the capacity of somatic mutation theory and mitochondrial metabolic theory to explain the origin of cancer. 

Response: We thank the reviewer for this comment.  We have now added Warburg’s quote to the end of manuscript and have commented on the implications of this quote to the take-home message in our concept paper.

Round 2

Reviewer 1 Report

I thank the authors for their time and efforts in revising the manuscript and addressing my concerns. 

I am fully aware that mitochondria, as a master metabolic regulator, play a critical role in cancer. Our research group demonstrated the mitochondrial metabolome of the cancer cells and expected subsequent groups to pay more attention to it rather than analyzing "bulk" cells/tissues only.

However, I am still not convinced that the proposed theory or any single theory could reflect the actual processes that happened in the oncogenesis and related events. There would be some support and opposing evidence. Our knowledge would be incomplete, and we are biased. Furthermore, a major number of experimental reports are not reproducible. Hence, we need more experimental evidence and (later) a quantitative model to assist the interpretation.

Regarding the therapeutic opportunities (chemotherapy, radiation therapy, and immunotherapy), more evidence on the success of newly developed therapies (based on our understanding of the metabolic disturbances of cancer cells and their surrounding environment) is required. There are available concerns that mitochondria-targeting agents may not work well.

Finally, I would like the authors to consider using "soft" sentences when discussing matters that would trigger unnecessary debates or negative impressions (e.g., the paragraph discussing Otto Warburg and his contribution).
